# Deep Learning-Based Method for Classification of Sugarcane Varieties

**Priscila Marques Kai** [1,*], **Bruna Mendes de Oliveira** [2] and **Ronaldo Martins da Costa** [1]

1   Institute of Informatics, Federal University of Goias, Goiânia 74690-900, Brazil
2   Agronomy School, Federal University of Goias, Goiânia 74690-900, Brazil
*   Correspondence: priscila.kai@hotmail.com

**Abstract:** The classification of sugarcane varieties using products derived from remote sensing allows for the monitoring of plants with different profiles without necessarily having physical contact with the study objects. However, differentiating between varieties can be challenging due to the similarity of the spectral characteristics of each crop. Thus, this study aimed to classify four sugarcane varieties through deep neural networks, subsequently comparing the results with traditional machine learning techniques. In order to provide more data as input for the classification models, along with the multi-band values of the pixels and vegetation indices, other information can be obtained from the sensor bands through RGB combinations by reconciling different bands so as to yield the characteristics of crop varieties. The methodology created to discriminate sugarcane varieties consisted of a dense neural network, with the number of hidden layers determined by the greedy layer-wise method and multiples of four neurons in each layer; additionally, a 5-fold evaluation in the training data was composed of Sentinel-2 band data, vegetation indices, and RGB combinations. Comparing the results acquired from each model with the hyperparameters selected by Bayesian optimisation, except for the neural network with manually defined parameters, it was possible to observe a greater precision of 99.55% in the SVM model, followed by the neural network developed by the study, random forests, and kNN. However, the final neural network model prediction resulted in the 99.48% accuracy of a six-hidden-layers network, demonstrating the potential of using neural networks in classification. Among the characteristics that contributed the most to the classification, the chlorophyll-sensitive bands, especially B6, B7, B11, and some RGB combinations, had the most impact on the correct classification of samples by the neural network model. Thus, the regions encompassing the near-infrared and shortwave infrared regions proved to be suitable for the discrimination of sugarcane varieties.

**Keywords:** machine learning; precision agriculture; remote sensing; Sentinel-2; sugarcane

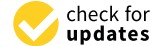

## 1. Introduction

With various products derived from sugarcane, such as food, fuel, alcoholic beverages, and plant biomass, it is considered a crop of extreme economic importance in countries such as Brazil and India, which together produce more than 50% of the world's total supply. In Brazil, the estimated cultivated area destined for sugarcane in the 2022/2023 harvest comprises approximately 8127.7 thousand hectares according to Conab [1].

Given the importance of the crop, research institutions have focused on the genetic improvement of sugarcane, developing new varieties aimed at enhancing the performance of the sugarcane sector, financed by mills, sugarcane suppliers, and distilleries [2]. Furthermore, in Brazil, the Cultivar Protection Law guarantees plant creators' right to ownership of cultivars, thus encouraging research [3].

Due to the diversity of sugarcane varieties, it is possible to find plants with different profile types—from general aspects, agricultural productivity, growth time, up to management recommendations, among other points—according to the characteristics of each

variant [2]. Thus, sugarcane producers can choose varieties that can best adapt to the climate, such as those with more disease resistance or sucrose production.

In order to monitor sugarcane crops, several studies have applied remote sensing techniques to allow for information collection without necessarily having physical contact with the objects of study, therefore reducing travel costs to the planting location. Part of the studies includes using satellite sensor images, portable sensor devices at the ground level, or sensors in the laboratory such as spectroradiometers or spectrometers, which are capable of measuring the amount of reflected radiation and irradiance at different wavelengths [4].

One advantage of satellite sensors is their periodicity, which makes it possible to obtain temporal data from regions in a time interval. Thus, acquiring data over several days from a locality according to the chosen product is admissible. Satellite sensor image databases provide some remote sensing products for free, such as the EOS LandViewer platform and the USGS Earth Explorer. These intuitive interfaces allow us to obtain sensor images of the earth's surface, such as through the use of Sentinel-2, Landsat-7, Landsat-8, MODIS, and other products, upon registration on the platform.

These satellite sensor images are obtained with various wavelengths, which are called bands. Thus, depending on the sensor used in the experiments, different sensing products may have varying amounts of bands. However, unlike the data captured in a controlled environment, satellite sensor data require attention to elements that can be unfavourable for the further analyses of images, such as clouds or shadows over the analysed terrain, which require image processing. For this reason, the selection of image data taken on days with favourable weather conditions helps to build a less noisy database.

For multispectral data, images of Landsat products (Landsat-7 and Landsat-8) [5,6], and more recently those of Sentinel-2 [7,8], are often used in the classification of sugarcane varieties. For hyperspectral images, there are articles on the application of EO1-Hyperion sensor images [9–12] and the data of spectroradiometer devices [13,14]. Since many of the articles related to the classification of sugarcane varieties have been published for more than ten years, some of the sensors used in those studies, such as Landsat-7 and Hyperion, are no longer in operation, and there is a need to look for other sensor options that could capture more recent images. Considering studies in which the database is composed of images, the pixel values acquired from the satellite sensor images of each variety are commonly used as input in classificatory models [8,10–12,15]. However, the availability of sensor images from regions with growing crops is limited by the occurrence of clouds at certain times of the year [16]. For this reason, many studies found in the literature make use of a single image in the classification of varieties [6,10–12,17].

Vegetation indices (VIs) are also included as input information in order to highlight vegetation conditions, thus helping to differentiate varieties [9,11,12]. Among the most applied vegetation indices in the literature is the Normalised Difference Vegetation Index (NDVI), which is important in providing information for analysing the condition of the crop and the plant nutrition, as seen in the articles [6,8,11,13].

In addition to the sensor's bands and vegetation indices, we can obtain more characteristics of the plants through RGB combinations. When we combine three bands, we have a coloured image with different proportions of red, green, and blue for each pixel of the resulting image. Thus, a natural RGB image represents an image composed of red, green, and blue bands of the visible spectrum, which is similar to how the human eye sees. In the case of a false colour image, we have band rearrangements that include non-visible wavelengths in order to emphasise or reveal information that is not available in the traditional configuration [18].

When approaching the discriminatory models applied in the classification of sugarcane varieties using satellite sensor images, there are articles in the literature that apply discriminant analysis, such as the studies by Galvão [11,12], which used images from Hyperion, and a study by Fortes [6], which employed Landsat-7 images. In addition, there are methods using spectral analysis, such as the Spectral Angle Mapper (SAM) [17], as well as supervised learning methods, which include the Random Forest applied by Duft to [7] and the Support

Vector Machines (SVM) [8,10] that, according to the literature on the classification of pixels of sugarcane varieties, have better precision in separating sugarcane varieties as compared to others classical algorithms, as seen in the studies by Everingham [10] and Kai [8].

However, when we include sugarcane monitoring [19] and the classification of crop varieties in general, artificial neural networks [20] and convolutional [21,22] ones are present in several recent articles, pointing to the relevance of neural networks in classifying cultures and their varieties. Nevertheless, these methods' potential in classifying sugarcane varieties still needs to be investigated, given the success achieved in other cultures.

For this reason, this article presents a methodology for classifying different sugarcane varieties that applies artificial neural networks, explores their potential for discrimination, and compares the results with methods usually employed in the differentiation task. In addition, Sentinel-2 images, with dates selected from between 2019 and 2020, were used to allow for more variability in the data related to sugarcane varieties. Furthermore, the data of simple RGB combinations from Sentinel-2 bands will also serve as input for the models to investigate these combinations' potential in discerning different sugarcane varieties.

## 2. Materials and Methods

This section will explore the materials and methods employed to develop the methodology for classifying sugarcane varieties with the use of deep neural networks through machine learning and remote sensing techniques.

### 2.1. Sugarcane Varieties

In this study, four sugarcane varieties from regions located in Goiás, Brazil were selected. Among the chosen varieties, RB867515, RB92579, and RB966928 correspond to the three most produced variants in the country according to the 2019/2020 harvest [2]. Among the characteristics of each sugarcane variety, we can highlight the following:

- RB867515: This represents a fast-developing variety with an upright growth habit, suitable for planting in environments of medium natural fertility. It has high sucrose content and agricultural productivity, with good performance in sandy textured soils.
- RB92579: This is a variety of slow development, recommended for planting in flat topography areas, floodplains, slopes, and plains. It has high agricultural productivity and rapid recovery from water stress.
- RB966928: This variety has medium sucrose content, high agricultural production, and early-to-medium maturation with high disease tolerance.
- RB988082: This is a sugarcane variety with a wide adaptability to environments, high agricultural productivity, medium sucrose content, and is harvested in mid-July.

The sugarcane harvest cycle of the four varieties covered in the study is available for visualisation in Figure 1.

| Varieties | Jan | Feb | Mar | Apr | May | Jun | Jul | Aug | Sep | Oct | Nov | Dec |
|-----------|-----|-----|-----|-----|-----|-----|-----|-----|-----|-----|-----|-----|
| RB867515  |     |     |     |     |     |     | ███ | ███ | ███ |     |     |     |
| RB92579   | ███ |     |     |     |     |     |     |     |     | ███ | ███ | ███ |
| RB966928  |     |     |     | ███ | ███ |     |     |     |     |     |     |     |
| RB988082  |     |     |     |     |     |     | ███ | ███ | ███ |     |     |     |

**Figure 1.** Sugarcane harvest cycle of varieties RB867515, RB92579, RB966928, and RB988082.

### 2.2. Images

For the experiments, orthorectified images from Sentinel-2, level 2A, and UTM/WGS84 projection were used, with pixel values provided in Bottom of Atmosphere (BOA); the images were composed of thirteen bands, with resolutions between 10 m and 60 m. The acquired images in the study were taken from the *EOS LandViewer* platform, with the selected sites containing four varieties of sugarcane located in the state of Goias.

The coordinates of each location were gathered by a portable GPS, with the subsequent conversion to a shapefile using the QGIS software to adapt the georeferenced data to the same coordinate system of Sentinel-2 images. The implementation of all codes, including the selection of the region of interest (ROI) as well as the training and testing of classificatory models, was developed in the Python language by the Spyder integrated development environment.

A total of 2 locations for each variety were selected, which resulted in 5 images of different dates per region and 10 images per variety between 2019 and 2020 (Table 1); these dates encompass the growth and maturation phases.

**Table 1.** Collection dates for the experiments.

| Date | Sigle | Date | Sigle | Sugarcane Variety |
|---|---|---|---|---|
| (18 March, 2 April, 27 April, 2 May, 22 May), 2020 | 22KEG\|22KEF | (8 May), 2019; (18 March, 2 April, 12 May, 27 May), 2020 | 22LGJ | RB988082 |
| (6 March, 10 April, 10 May, 30 May, 14 June), 2020 | 22KCE | (2 April, 2 May, 22 May, 1 June, 6 July), 2020 | 22LGH | RB867515 |
| (5 February, 26 April, 6 May, 21 May, 20 June), 2019 | 22KCE | (2 April, 27 April, 12 May, 27 May, 16 June), 2020 | 22KFF | RB92579 |
| (16 March, 31 March, 20 April, 10 May, 30 May), 2020 | 22KCF | (2 April, 12 May, 27 May, 11 June, 16 July), 2020 | 22KFF | RB966928 |

Visually, the occurrence of clouds and shadows was verified on each chosen date to rule out locations with direct interference from these factors. Due to adverse weather conditions and the sugarcane harvest period, data for some varieties had to be obtained in different years.

Due to each folder's considerable size (approximately 1.7 GB per file containing all Sentinel-e bands in separate images), rectangles that encompassed only the studied regions were manually selected on the online platform for download in order to streamline data processing and extraction.

Once the sensor images were acquired, different bands were combined to form simple RGB compositions. Vegetation indices and values representing each sensor band formed the research database, separating each sample by values contained in the pixels.

In this research, the database was composed of 20,000 samples,which correspond to the image pixels acquired by the EOS LandViewer platform (accessible through the link https://eos.com/landviewer/, accessed on 18 September 2022), with 5000 pixels per variety of sugarcane. Of these pixels, 80% were used in the training samples and 20% were used for testing.

### 2.3. Database

The experiments used three different pieces of information: the values of each band of Sentinel-2, ten vegetation indices, and eleven combinations of the sensor bands based on the Sentinel Hub repository [23]. Except for band 10 (Cirrus), all other bands of Sentinel-2 were included as input data to determine which ones were the most significant in distinguishing between sugarcane varieties. Each item of data used is available in Figure 2.

Furthermore, when working with vegetation indices, an adjustment of the spatial resolution applied in the calculation is required. Traditionally, the resampling methods employed are the nearest neighbour, bilinear, cubic, Lanczos, and Gauss. The bilinear method was selected to resample the Sentinel-2 band images for this work.

For the experiments, we made use of indices that are widely applied in the literature (Table 2), such as the Normalised Difference Vegetation Index (NDVI), Green Normalised Difference Vegetation Index (GNDVI), Normalised Difference Water Index (NDWI), Normalised Difference Chlorophyll Index (NDCI), Normalised Difference Moisture Index (NDMI), Enhanced Vegetation Index (EVI), Normalised Difference Red Edge Index (NDRE),

Soil-Adjusted Vegetation Index (SAVI) [6,13], Structure Insensitive Pigment Index (SIPI), and the Coloration Index (CI).

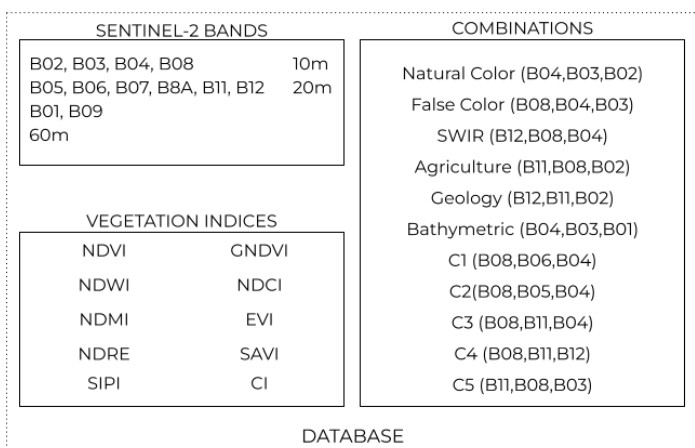

**Figure 2.** Database for the experiments. Each sample is composed of values from each Sentinel-2 band (except for band 10), vegetation indices, and combinations of the Sentinel-2 bands.

**Table 2.** Vegetation indices were used in the studies to highlight the differences between sugarcane varieties.

| Vegetation Index | Formula | Reference | Applications/Biological Parameter |
|---|---|---|---|
| CI | $\dfrac{\text{Red} - \text{Blue}}{\text{Red}}$ | [24] | Vegetation |
| EVI | $G \dfrac{(\text{NIR} - \text{Red})}{(\text{NIR} + C_1 \text{Red} - C_2 \text{Blue} + L)}$ | [25] | Green biomass |
| GNDVI | $\dfrac{\text{NIR} - \text{Green}}{\text{NIR} + \text{Green}}$ | [26] | Chlorophyll |
| NDCI | $\dfrac{\text{Rededge} - \text{Red}}{\text{Rededge} + \text{Red}}$ | [27] | Chlorophyll |
| NDRE | $\dfrac{\text{NIR} - \text{Rededge}}{\text{NIR} + \text{Rededge}}$ | [28] | Green biomass |
| NDMI | $\dfrac{820\,\text{nm} - 1600\,\text{nm}}{820\,\text{nm} + 1600\,\text{nm}}$ | [29] | Water content |
| NDVI | $\dfrac{\text{NIR} - \text{Red}}{\text{NIR} + \text{Red}}$ | [30] | Green biomass |
| NDWI | $\dfrac{\text{NIR} - \text{SWIR}}{\text{NIR} + \text{SWIR}}$ | [31] | Water content |
| SAVI | $\dfrac{(1 + L)(\text{NIR} - \text{Red})}{\text{NIR} + \text{Red} + L}$ | [32] | Green biomass |
| SIPI | $\dfrac{800\,\text{nm} - \text{R445}\,\text{nm}}{800\,\text{nm} + 680\,\text{nm}}$ | [33] | Chlorophyll |

As the bands derived from Sentinel-2 images were considered as one of the input characteristics of the pixels, in order to avoid the repetition of information associated with each pixel, each RGB combination acquired was converted into grey levels to obtain different tonalities from the studied region. The combinations containing 3 bands were converted into greyscale images, considering the standard RGB conversion (0.2989 × R + 0.5870 × G + 0.1140 × B).

In addition to sensor band values, vegetation indices, and RGB combinations, tests were separately performed for each piece of information in order to analyse its potential for data differentiation in the classification of sugarcane varieties.

## 2.4. Classification Models

Among the evaluated classification models, our study developed the k-Nearest Neighbours algorithm, Support Vector Machines (SVM), Random Forests (RF), and the Dense Neural Network. The characteristics of each classifier can be seen in Table 3, which contains the advantages and disadvantages of each model.

**Table 3.** Advantages and disadvantages of the discriminatory models.

| Classifier | Advantages | Disadvantages |
| --- | --- | --- |
| kNN | Simple and easy to implement algorithm; can be used in regression problems. | Not as efficient for data with high dimensionality; a balanced database is required; becomes slower as variables increase and is sensitive to outliers. |
| Random Forest | Not sensitive to outliers; it can also be used in regression problems; little risk of overfitting; good efficiency for large data sets. | Slow training; they are not easy to interpret; computationally intensive. |
| Support Vector Machine | Effective for data with high dimensionality, kernel flexibility, and proper functioning, with clear separation margin. | Not suitable for large databases and noisy data. |
| Artificial Neural Networks | Continuous Learning; flexible in regression and classification problems. | Slow training; hardware-dependent; complex algorithms; efficiency compromised with small and noisy databases. |

Convolutional Neural Networks were not applied in the study due to the size of some regions. When considering the size per pixel of the Sentinel-2 product, it was impossible to develop a deeper network due to the size limitation, which resulted in the availability of few samples that could serve as the model entry.

### 2.4.1. Hyperparameter Adjustment

In order to adjust the hyperparameters for each model, Bayesian optimisation was applied. Unlike other existing techniques such as the Grid Search method, which tests all previously fixed combinations, and the Random Search method, which defines values between intervals to be tested randomly at each iteration, Bayesian optimisation allows for a faster hyperparameter search that is based on previously performed evaluations.

The choice of parameters for the models was based on the highest precision obtained after the runs for each parameter. For example, among the parameters evaluated in the model using k-NN are the *leaf_size*, *n_neighbours* (number of neighbours) and p (metric distance). The range defined for the possible values of *k* was between 1 and 30 neighbours. Each evaluated parameter can be seen in Table 4.

**Table 4.** Models evaluated in the study and parameters tested.

| Model | Parameter |
| --- | --- |
| Artificial Neural Network | loss (categorical_crossentropy, mean_squared_error) batch_size (5, 15, 30), learning_rate (0.001,1) |
| k-NN | n_neighbors (1:30), p(1,2) |
| Random Forest | n_estimators (50, 100, 150, 200, 250, 300) max_features (auto, sqrt), max_depth (2:9) bootstrap (true,false) |
| SVM | kernel (linear, polynomial, sigmoid, rbf) C (0.01, 0.1, 1, 10, 100, 1000) |

### 2.4.2. Construction of the Dense Neural Network

The greedy pre-training method (greedy layer-wise) was used to define the number of hidden layers. The model was adjusted by each layer added, thus training shallow sub-networks [34]. Consequently, the model learned the inputs that made up the layer. In this way, we had the training in parts at each iteration, therefore making the training process for the network less onerous. The configuration was chosen by the accuracy of the training and test data obtained by inserting each layer.

In this study, ten iterations of the hidden-layer addition were defined, evaluating the structure's performance per additional layer and graphically comparing the results between the number of layers.

For each addition, a predetermined amount of neurons were added to the new hidden layer in order to check for accuracy. If the inputs were formed only by vegetation indices, we used ten neurons per hidden layer, eleven with RGB combinations, twelve with Sentinel-2 bands, and thirty-three with the union of all the data. In this process, the training was composed of 500 epochs in conjunction with the "*EarlyStopping*" callback, patience was equal to 10 (number of epochs with no improvement), and the average accuracy was obtained by evaluating the use of a 10-fold cross validation.

After selecting the number of hidden layers, the parameters related to the number of neurons present in each hidden layer were defined, and the neural network training was optimised. Testing various structural configurations with the number of neurons allowed us to assess how the increase or decrease in neurons in each layer influenced the separation between classes.

### 2.5. Accuracy Assessment

After the selection of hyperparameters for each model tested in this study, by cross-validating five times and applying the Bayesian optimisation technique to measure the performance of the models, the database was then divided into samples for the training set, 80% of which represented 16,000 pixels distributed among four sugarcane varieties, while 20% of the samples for the test set had 4000 in total. All data were normalised on a scale between 0 and 1.

The entire process carried out for the classification of sugarcane varieties can be seen in Figure 3.

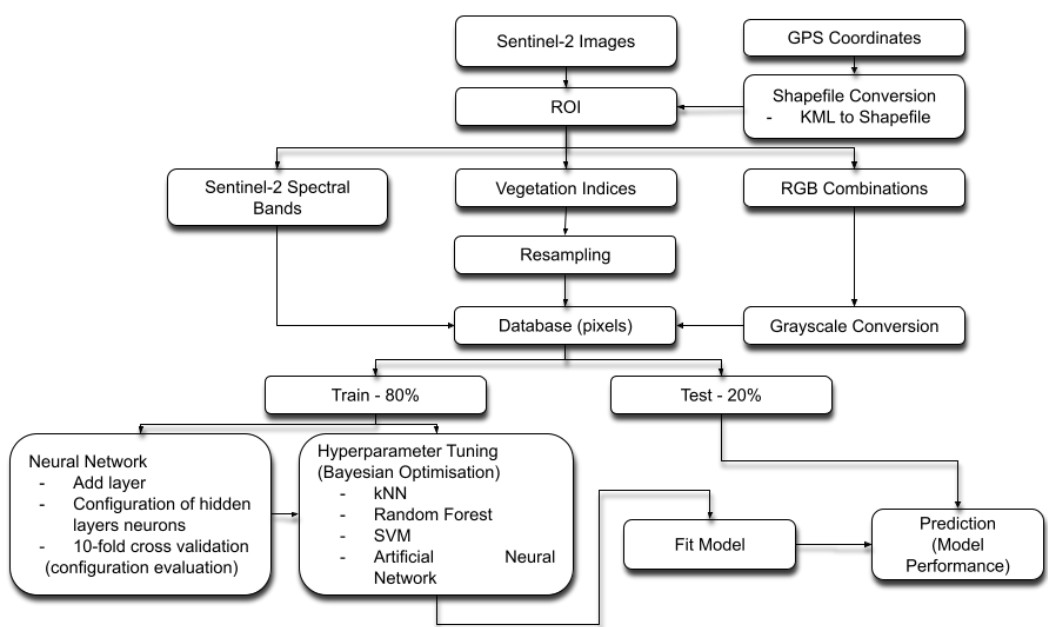

**Figure 3.** Flowchart for classifying sugarcane varieties.

Initially, we acquired Sentinel-2 images through the online platform, which were later masked through shapefile files generated from coordinates captured by a portable GPS. Next, the construction of the base was applied to the study, which was formed by three different types of information; all had been derived directly from the sensor. These data were divided into training samples, which served for hyperparameter adjustment and the training of the final model, and test samples, which were used to evaluate the model.

In order to evaluate the contributions of each feature in the neural network prediction, the SHAP method (SHapley Additive exPlanations) was applied. With the SHAP values, we could analyse the impact and interaction between the features used as input in the [35] model.

## 3. Results and Discussion

This section demonstrates the results of experiments aimed at classifying four different sugarcane varieties. In addition to the traditional machine learning techniques evaluated in the study, a dense neural network developed during the experiments is presented and compared with the other methods.

### 3.1. Classification of Sugarcane Varieties by Artificial Neural Networks

Through pre-training, the samples belonging to the training set were applied to determine the number of hidden layers in the neural network for classifying sugarcane varieties, subsequently evaluating the network's performance at each inserted layer. The number of iterations performed added up to 11 hidden layers, with the initial run having 2 layers (1 hidden layer plus the output layer) and 10 additional iterations, resulting in a total of 12 network layers.

In Figure 4, we evaluate the model according to the number of epochs and total layers of the neural network. In this example, we have all the hidden layers, with the number of neurons being equal to the number representing the model inputs.

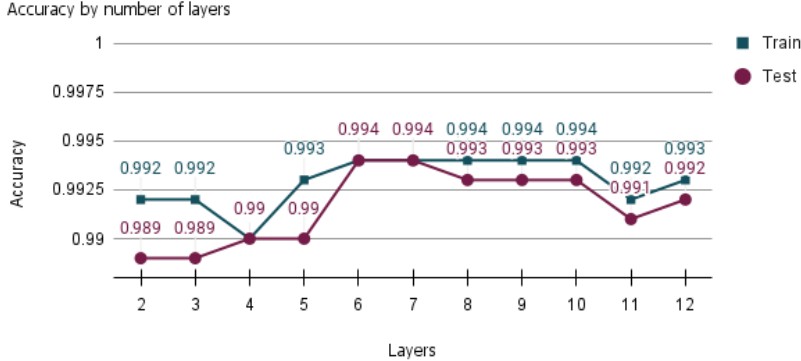

**Figure 4.** Evaluation of the dense neural network model according to the number of layers added.

In this network structure with the same number of neurons in all hidden layers, the accuracy improved up to the fifth hidden layer of the neural network model; thus, there was a total of six layers. Because it is a non-linear, separable problem containing a lot of data that overlap in between samples, a neural network with a single hidden layer did not achieve the same efficiency as compared to other configurations with more layers.

The drop in model performance at some points may mean overfitting the data by inserting more hidden layers and increasing the accuracy of subsequent layers. For this reason, the six-layer neural network structure was selected for the hyperparameter adjustment. More hidden layers did not improve the model's discernment in separating sugarcane varieties. To illustrate the arrangement of neurons in each layer, we have the number of inputs (33 inputs, composed of Sentinel-2 bands, vegetation indices, and RGB combinations), the neurons of each hidden layer, and the outputs.

The Bayesian optimisation applied to the database with all 33 features to define the number of neurons in the hidden layers did not obtain the best result among the configuration of neurons tested (configuration (33,63,11,7,64,64,4) with learning rate = 0.001 and batch size = 5). For this reason, the definition of the number of neurons in each hidden layer considered some existing methods; for example, values between the size of the input and output layers [36] were considered (configuration with the number of inputs plus the number of neurons in each hidden layer and the output for six layers). Multiple values of 4 were also tested, with a maximum of 128 neurons in the layer closest to the input.

For networks of six layers, with the first hidden layer composed of 64 neurons, an accuracy above 99.2% was verified for the training dataset in addition to the better precision of the options with fewer neurons. Adding more neurons in hidden layers did not improve the model's accuracy in predicting sugarcane varieties (Table 5).

**Table 5.** Evaluation of neural network models for different numbers of neurons in the hidden layers.

| Features | Configuration | Accuracy (Train Data) |
|---|---|---|
| Bands, Vegetation Indices, and RGB Combinations | (33,19,12,8,6,5,4) | 98.4 (+− 0.50) |
| | (33,33,19,12,8,6,4) | 99.0 (+− 0.17) |
| | (33,63,11,7,64,64,4) with BO | 96.2 (+− 1.29) |
| | **(33,64,32,16,8,4,4)** | **99.2 (+− 0.19)** |
| | (33,128,64,32,16,8,4) | 99.1 (+− 0.50) |
| Bands | (12,8,6,5,4) | 96.1 (+− 2.14) |
| | (12,12,8,6,4) | 98.0 (+− 0.74) |
| | (12,32,16,8,4) | 98.9 (+− 0.39) |
| | (12,64,32,16,4) | 98.2 (+− 0.39) |
| Vegetation Indices | (10,10,7,6,5,5,4) | 89.9 (+− 1.16) |
| | (10,32,16,8,4,4,4) | 94.6 (+− 1.07) |
| | (10,64,32,16,8,4,4,4) | 96.1 (+− 0.57) |
| RGB Combinations | (11,11,8,6,4) | 94.1 (+− 2.25) |
| | (11,16,8,4,4) | 96.0 (+− 0.62) |
| | (11,32,16,8,4) | 97.3 (+− 0.22) |
| | (11,64,32,16,4) | 98.2 (+− 0.39) |

There were no improvements in the network with the dropout between layers, so it was not used in the prediction. To update the weights and biases, the optimiser chosen was Adam, which had a learning rate equal to 0.001 and a categorical cross-entropy for the loss function.

An analysis of the classification of varieties using only the values contained in the Sentinel-2 bands showed that the constructed network resulting from the process of adding hidden layers was composed of four layers in total due to the non-improvement of the model after adding the sixth hidden layer (Figure 5).

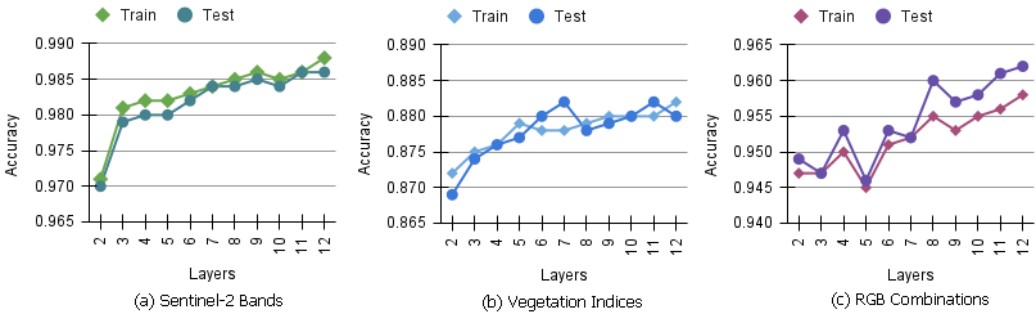

**Figure 5.** Evaluation of the dense neural network model according to the number of hidden layers added for bands, vegetation indices, and RGB combinations.

In comparing the structure formed with the network previously defined using all 33 features, the potential for differentiation between the four varieties proves to be remarkable,

with an accuracy of around 98%, which indicates that the sensor bands can provide a good classification among the classes.

The separability of varieties with the use of vegetation indices decreased the accuracy as compared to when pixel information from Sentinel-2 bands was used. The decrease may have several factors, such as drought, spectral mixing, and diseases, which can affect the reading of vegetation conditions and confuse the separation between sugarcane varieties.

The number of hidden layers in the network was defined as being six hidden layers that resulted from the decrease in accuracy of the validation and oscillation during the iteration of more layers. In order to improve the results, using more vegetation indices as the characteristics of varieties can help to increase the discrimination between the classes if the input is only composed of those indices.

In applying Sentinel-2 bands to form the simple RGB combinations, the neural network evaluation obtained results that were superior to the vegetation indices, with an accuracy of 98.2% by the average of each fold, and with the first hidden layer containing 64 neurons. This fact can be explained based on the good discrimination between varieties by using only the sensor band data, which contributed to the results of the combinations.

### 3.2. Machine Learning Models in the Classification of Sugarcane Varieties

After returning the hyperparameters of the kNN, Random Forest, and Support Vector Machine algorithms via the Bayesian Optimisation process (Table 6), each model was trained using all the samples belonging to the training set, and a performance evaluation was conducted using test data composed of 4000 samples.

**Table 6.** Selected parameters and accuracy of each model.

| Model | Hyperparameter |
|---|---|
| kNN | n_neighbors: 1, p: 2 |
| Random Forest | max_features: 0.62, max_samples: 0.99, n_estimators: 139 |
| SVM | C: 832.6 |
| ANN | batch_size: none, loss: categorical_crossentropy, learning_rate: 0.001 |

In the case of neural networks, the results derived from Bayesian optimisation were not satisfactory. Thus, dimensionality reduction was thought to be a solution that could enable the application of Bayesian optimisation to define the number of neurons in the hidden layers. However, it was possible to verify the potential of deep neural networks in separating sugarcane varieties with an accuracy of above 99%. More investigations regarding the definition of the ideal number of neurons in the hidden layers are necessary to improve the model's accuracy.

With the application of Bayesian optimisation, the support vector machines showed efficiency in separating the four varieties to fit the hyperparameters of each model (Table 7), which had the highest accuracy among the methods tested (99.55%).

An analysis of the classification of each sugarcane variety showed that the RB92579 obtained the highest number of incorrect predictions using kNN and Random Forest, and it is often confused with the RB966928 variety. The possible causes for misclassification may be related to the spectral mixture of the growing variety with the exposed soil. Due to the slow growth characteristic, acquiring images in the months close to the harvest season, namely between October and January, may have impacted the classification models' correct separation of the samples.

In the case of the neural network and the SVM results, the producer's accuracy in relation to varieties is more distributed among the classes, with a more significant proportion of samples being correctly classified according to the predicted class. Consequently, user's accuracy is also impacted, resulting in fewer false positives. In the case of the RB988082 variety, the SVM was able to correctly classify all variety samples.

Evaluating the contribution of each characteristic used as input in the dense neural network constructed from the SHAP values in the study, we can visualise its impact on the correct classification of each sugarcane variety.

In Figure 6, it is possible to verify that the Sentinel-2 bands are the most critical characteristics in the differentiation between the four sugarcane varieties, especially bands B1 and B6 as well as between B7 and B11, which has greater relevance for the classification. A similar result was observed using SVM (Figure 7), which also points to B6 as being the most significant feature for correctly classifying the studied sugarcane varieties, mainly for the RB988082 variety (Class 0).

**Table 7.** Overall accuracy, user's accuracy, and producer's accuracy by model.

| kNN | | Reference data | | | | | User's accuracy |
|---|---|---|---|---|---|---|---|
| | | 0 | 1 | 2 | 3 | Total | |
| Classified Data | 0 | **991** | 0 | 1 | 0 | 992 | 99.90% |
| | 1 | 1 | **1015** | 10 | 4 | 1030 | 98.54% |
| | 2 | 2 | 6 | **969** | 6 | 983 | 95.57% |
| | 3 | 1 | 3 | 24 | **967** | 995 | 97.18% |
| | Total | 995 | 1024 | 1004 | 977 | **4000** | **OA 98.55%** |
| Producer's accuracy | | 99.60% | 99.12% | 96.51% | 98.98% | | |

| Random Forest | | Reference data | | | | | User's accuracy |
|---|---|---|---|---|---|---|---|
| | | 0 | 1 | 2 | 3 | Total | |
| Classified Data | 0 | **992** | 0 | 0 | 0 | 992 | 100.00% |
| | 1 | 1 | **1020** | 5 | 1 | 1027 | 99.32% |
| | 2 | 2 | 1 | **985** | 2 | 990 | 99.49% |
| | 3 | 0 | 3 | 14 | **974** | 991 | 98.28% |
| | Total | 995 | 1024 | 1004 | 977 | **4000** | **OA 99.28%** |
| Producer's accuracy | | 99.70% | 99.61% | 98.11% | 99.69% | | |

| SVM | | Reference data | | | | | User's accuracy |
|---|---|---|---|---|---|---|---|
| | | 0 | 1 | 2 | 3 | Total | |
| Classified Data | 0 | **995** | 0 | 0 | 0 | 995 | 100.00% |
| | 1 | 0 | **1022** | 6 | 3 | 1031 | 99.13% |
| | 2 | 0 | 0 | **993** | 2 | 995 | 99.80% |
| | 3 | 0 | 2 | 5 | **972** | 979 | 99.28% |
| | Total | 995 | 1024 | 1004 | 977 | **4000** | **OA 99.55%** |
| Producer's accuracy | | 100.00% | 99.80% | 98.90% | 99.49% | | |

| ANN | | Reference data | | | | | User's accuracy |
|---|---|---|---|---|---|---|---|
| | | 0 | 1 | 2 | 3 | Total | |
| Classified Data | 0 | **989** | 0 | 0 | 0 | 989 | 100.00% |
| | 1 | 0 | **1022** | 4 | 2 | 1028 | 99.42% |
| | 2 | 0 | 2 | **997** | 4 | 1003 | 99.40% |
| | 3 | 6 | 0 | 3 | **971** | 980 | 99.08% |
| | Total | 995 | 1024 | 1004 | 977 | **4000** | **OA 99.48%** |
| Producer's accuracy | | 99.40% | 99.80% | 99.30% | 99.39% | | |

Note: Class 0 = RB988082, Class 1 = RB867515, Class 3 = RB92579, and Class 4 = RB966928.

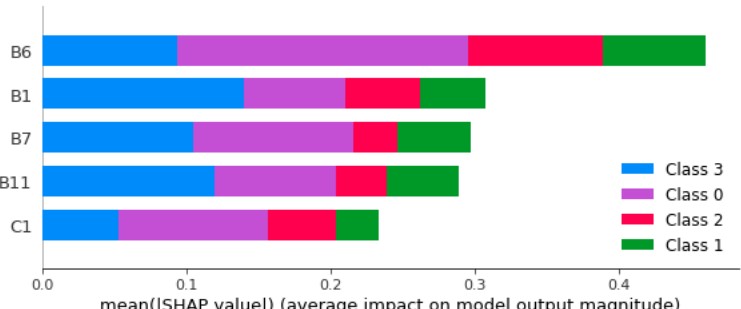

**Figure 6.** The five most significant features as input in the neural network model for the correct classification of sugarcane varieties. Among the characteristics, the bands B6 (band 6—Vegetation Red Edge), B1 (band 1—Coastal Aerosol), B7 (band 7—Vegetation Red-Edge), B11 (band 11—SWIR), and combination C1 (band 8, band 6, and band 4).

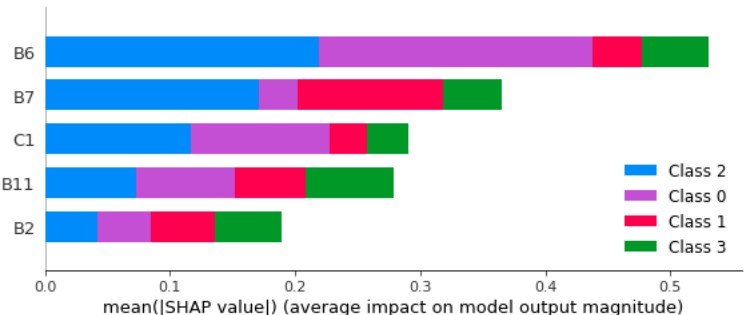

**Figure 7.** The five most significant features as input in the SVM model for the correct classification of sugarcane varieties. Respectively, the Sentinel-2 bands B6 (band 6—Vegetation Red-Edge), B7 (band 7—Vegetation Red-Edge), combination C1 (band 8, band 6, and band 4), B11 (band 11—SWIR), and B2 (band 2—Blue).

The contribution by bands to the correct classification of sugarcane varieties, such as that by bands 6 and 7, can be associated with their common application to vegetation classification, with the red-edge region being more sensitive to chlorophyll levels in the leaves [37] and presenting different values for each variety, thus helping in the separation. Band 2 (blue) presents the sensitivity to carotenoids with high chlorophyll absorption, favouring plant differentiation and providing a good contrast between different vegetation types by band 11.

For the band combination, the C1 combination (Band 8, Band 6, Band 4) is composed of chlorophyll-sensitive bands and encompasses regions in red such as band 4, red-edge, and near-infrared (NIR), thus highlighting regions of dense vegetation and making it possible to obtain different reflectance values according to chlorophyll absorption for each variety.

Among the RGB combinations, combination C1 (B08, B06, B04) by the ANN model, and False Colour (B08, B04, B03) as the third most significant characteristic that resulted from the Random Forest model (B6, B1, False Colour, B11, and C1, respectively), are also among the five most striking features that encompass the red, near-infrared, and shortwave infrared regions, demonstrating that these regions of the spectrum, especially in the near-infrared region, are suitable for sugarcane discrimination, as reported in the literature [6].

As for vegetation indices, their lower precision in comparison to the other characteristics used as input in the models may have resulted from the type of vegetation index used in the study, which is not the most effective in separating varieties. Testing other vegetation indices is a way of searching for more efficient indices that would allow for an extraction of data that could favour the separation between varieties, thus increasing classification accuracy by using a wide range of vegetation indices and selecting those most sensitive to different biochemical contents.

In addition, other factors may have interfered in the resulting calculation of the indices, impairing the classification of the models, such as soil conditions, exposed soil, spectral mixture, diseases and pests, application of biochemicals, image acquisition on dates close to the harvest period, or water stress. Since some varieties are more tolerant than others in some aspects, the calculation performed may not be an accurate representation of the variety that is less tolerant to any of the above-mentioned factors.

Furthermore, each variety's cycle must be considered in order to select the most favourable dates for analysis, preferably with a closed canopy.

## 4. Conclusions

This study presented a methodology based on neural networks for classifying four Brazilian sugarcane varieties. It added RGB combinations to the models as network input values, as well as spectral bands and vegetation indices, allowing us to investigate the discriminatory potential of the inserted characteristics as an input for the model. Among the obtained results, the classification of the network applying all the extracted characteristics obtained an accuracy of 99.48%, demonstrating its potential in the differentiation between sugarcane varieties.

Using neural networks in the classification of the most significant characteristics of the varieties, the Sentinel-2 bands proved to be more efficient in separating the four sugarcane varieties, particularly chlorophyll-sensitive bands belonging to the red, near-infrared, and shortwave infrared regions, which were suitable for sugarcane discrimination. The RGB combinations also indicate the potential of combining different bands in order to enhance plant characteristics that could serve as input in classification models.

Despite the inferior accuracy resulting from vegetation indices, they are widely applied to several plant identification and classification problems, indicating that their application is relevant. However, more indices need to be explored in order to find more efficient indices for the analysis of biochemical content in sugarcane plants, allowing for the acquisition of information that could favour the separation between varieties. Furthermore, factors such as the occurrence of diseases, soil exposure causing spectral mixing, drought, and chemicals, among others, may have influenced the model's prediction.

**Author Contributions:** The conceptualisation, data extraction, tests, and writing were carried out by P.M.K., with the guidance of R.M.d.C. at the Institute of Informatics of the Federal University of Goias, and by B.M.d.O. at the School of Agronomy of the Federal University of Goias. All authors have read and agreed to the published version of the manuscript.

**Funding:** This study was funded in part by Higher Education Personnel Improvement Coordination–Brazil (Coordenação de Aperfeiçoamento de Pessoal de Nível Superior—CAPES), Finance Code 001.

**Data Availability Statement:** All Sentinel-2, Product 2A images applied in the survey can be obtained free from the EOS LandViewer online platform through the following link: https://eos.com/landviewer/ (accessed on 1 May 2022) (registration required).

**Acknowledgments:** This research was supported by LaMCAD/UFG and Ridesa partners by providing information on the sugarcane plantation sites used in this study.

**Conflicts of Interest:** The authors declare no conflict of interest.

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
