# Peer review of "Deep Learning-Based Method for Classification of Sugarcane Varieties"

_agronomy, doi:10.3390/agronomy12112722_

Round 1
Reviewer 1 Report
The manuscript has the aim to classify sugarcane varieties using Sentinel images and deep learning. The manuscript has an important application for agricultural analyses and could be applied to monitor sugarcane development. However, I have some comments about the paper structure, especially for introduction, methodology and results and discussion chapter. The introduction has not sufficient references to explain the problem and novelty of the study. The material and methods could be clearer and more concise. Moreover, the results and discussion are not good describe during the text, it would be good see some statistics that authors just comment during the text but not show well.
Specific comments:
-Abstract: Key-words are equal to title words; they need to be re-written. Abstract needs objectives, methodology and the results.
-Introduction
The introduction is poorly. Needs citation, examples and what is the new of this work.
The authors need to write about studies that used images to classify varieties, state of art. There are some studies? They have to show the novelty of their study.
Explain the deep learning, why? Write examples and advantages of this technique to classify sugarcane varieties.
-Materials and Methods
Why did the authors choose all bands of Sentinel-2?
What are the advantage of use Natural color and false color? The spectral behavior changes when used only the bands?
Line 124-125 –clarify
Add the differences between the varieties selected in this study. The average yield, type of soil, production environment. That could help the authors discus better the results.
-Results and Discussion
Line 249-251: This part is about methodology and not results. Rewrite.
The objective is to compare different machine learning and deep learning techniques to classify varieties? That is not clear in the methodology and objectives. Clarify in all the text.
Show the confusion matrix resulted of the validation, only the overall accuracy it is not sufficient. Discuss the results, the overall accuracy, the producer and user accuracies.
The discussion is poorly, try to explain better the results. What are the differences of this study when compare with others?
The argument about the vegetation index is poorly. Why more indices would be necessary? The authors need to explain and show the reasons clearly.
-Conclusion
Conclusion needs to re-write based on the suggestions about results and discussion
Author Response
Please see the pdf attachment containing the corrections made to the text.

Reviewer 2 Report
1. The author should discuss more relevant research work to provide theoretical basis for the research methods.
2. In order to highlight the contribution of this paper, in reviewing the literature, the author should discuss the work not done by previous researchers to clarify the development of the methods in paper.
3. The author should provide more information about the research data, such as image data.
4. Figure 2 is the core of the article, but it is not very clear and needs to be improved.
5. The author should provide more detailed data on the impact of different input features on the classification results, so that the results of machine learning methods and artificial neural networks are comparable.
6. There are some grammar and spelling mistakes, so the English level needs to be improved to improve the paper.
Author Response

(The authors gave the same response as above.)

Round 2
Reviewer 1 Report
The authors have made a good job to improve the paper.
The authors commented about the comparison with other machine learning algorithms. However, the abstract didn’t explain what are the best algorithm. Please, clarify the methodology of abstract.
Figures 6 and 7: describe in legend what is B11, B7, …. When explain in the text about the most important variables it is important use arguments about the spectral behavior. Add this discussion, because of water? Structure? Leaves? Biomass?
Author Response
Please see the attachment with the cover letter.
